# Simple Temporal Adaptation to Changing Label Sets: Hashtag Prediction via Dense KNN

**Niloofar Mireshghallah**[1][*], **Nikolai Vogler**[1][*], **Junxian He**[2]
**Omar Florez**[4][**], **Ahmed El-Kishky**[3][**], **Taylor Berg-Kirkpatrick**[1]
[1]University of California San Diego, [2]Carnegie Mellon University, [3]Twitter, [4]LatinX in AI
{fmireshg,nvogler,tberg}@ucsd.edu, junxianh@cs.cmu.edu
omarflorez.research@gmail.com, aelkishky@twitter.com

## Abstract

User-generated social media data is constantly changing as new trends influence online discussion and personal information is deleted due to privacy concerns. However, traditional NLP models rely on fixed training datasets, which means they are unable to adapt to temporal change—both test distribution shift and deleted training data—without frequent, costly re-training. In this paper, we study temporal adaptation through the task of longitudinal hashtag prediction and propose a non-parametric dense retrieval technique, which does not require re-training, as a simple but effective solution. In experiments on a newly collected, publicly available, year-long Twitter dataset exhibiting temporal distribution shift, our method improves by $64\%$ over the best static parametric baseline while avoiding costly gradient-based re-training. Our approach is also particularly well-suited to dynamically deleted user data in line with data privacy laws, with negligible computational cost/performance loss.

## 1 Introduction

Distribution shift, particularly in the target labels of classification tasks, presents a serious challenge for the deployment of NLP systems in real-world scenarios. The distribution of text data changes over time due to new language usage, events, or trends (Eisenstein et al., 2014; Ryskina et al., 2020; Jaidka et al., 2018), which causes statically-trained models to become stale without further re-training on new data (Lazaridou et al., 2021; Dhingra et al., 2022; Luu et al., 2022). Temporal shift is most evident in social media data, which we study through multi-label tweet-hashtag prediction, where tweets are classified into rapidly changing user-generated trends. Recent temporal analysis of named entity recognition models for social media (Rijhwani and

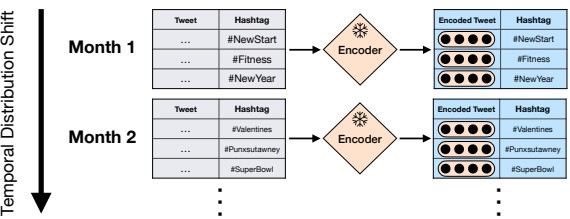

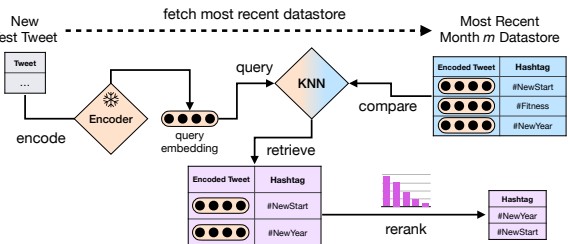

Figure 1: An overview of the KNN Dense Retrieval model components. First, a month's tweets are encoded into embeddings and added to an external, indexed datastore with their associated hashtags—a process that could be automated in a production setting. Second, a test tweet is encoded and used to query the most recent datastore, which contains the latest month's training data for temporal adaptation. The top-$k$ tweets from this datastore are then retrieved and re-ranked.

Preotiuc-Pietro, 2020) identifies the advantages of continual exposure to late-breaking training data. Given the ever-shifting nature of user-generated data, deployed models must be explicitly designed to (1) *adapt* to dynamic test distributions to prevent temporal performance degradation, and (2) *abide* by data privacy laws, such as GDPR, CCPA, etc. (Voigt and Von dem Bussche, 2017), that mandate prompt removal of user data.

In this paper, our main contribution is an empirical analysis of a known, but understudied learning paradigm in the context of temporal adaptation: non-parametric classification via dense retrieval from a datastore. By using a $K$-nearest neighbor (KNN) classifier in conjunction with a static neural text encoder and simple reranker (as depicted in Fig 1), we demonstrate that continual and automatic updates of the training datastore facilitate quick adaptation and deletion. In contrast to con-

---
[*]Equal contribution.
[**]Work done while at Twitter.

temporary neural paradigms, which require gradient descent during fine-tuning, our approach requires no gradient updates—both adaptation and retroactive deletion use minimal compute.

In order to analyze our proposed temporal adaptation method, we introduce a new supervised dataset (§3) focused on *hashtag prediction*, a multi-label classification task in the social media domain that exhibits label shift over time. Hashtag prediction, in which a system must predict the set of hashtags to be assigned to a given input tweet, is particularly amenable to our non-parametric method due to the intrinsic, user-generated categorization. In other words, the latest supervised training data can be automatically scraped and updated periodically on-demand. We release the dataset as tweet IDs.

In addition to comparing several text encoding strategies and re-ranking methods on the dataset (§5), we compare against two state-of-the-art parametric baselines based on BART (Lewis et al., 2020), a large pre-trained sequence-to-sequence model. These variants include a fine-tuned BART encoder-based classifier, along with the full BART seq2seq model fine-tuned on tweet-hashtag pairs. We find that our best non-parametric approach outperforms *static* parametric models with an average relative gain of $64\%$ recall under test distribution shift—and, even outperforms conventional gradient-based *temporal adaptation* of parametric baselines with an average relative gain of $12\%$ recall. Together, our empirical analyses highlight non-parametric techniques as a practical and promising direction for handling distribution shift and user-deletion, and may facilitate future work on real-world, temporal NLP system deployment.

## 2 Dense KNN Hashtag Prediction Model

Language in user-generated social media data constantly changes, which makes the prediction of new trend categories, or hashtags, challenging for NLP models. Fundamentally, hashtag prediction is a multi-label classification problem, in which a set of hashtags must be suggested for a single tweet. As new events influence online discussion, the classification labels change drastically: roughly half of hashtag types in our dataset are newly replaced after 4–5 months (see details in App. Fig. 3).

For this task, we consider a known, but understudied learning paradigm in the context of temporal adaptation: non-parametric classification via dense retrieval from a datastore. By combining a

$K$-nearest neighbor (KNN) classifier with a static neural text encoder and simple reranker, as depicted in Fig. 1, we enable simple but effective updating of an external datastore when train/test data changes, in contrast to full re-training of parametric models. Our model first retrieves the top-$K$ nearest hashtag labels from a datastore based on L2 distance in neural embedding space,

$$\mathcal{S}_k(x_t) = \{(x_1, y_1), \dots, (x_k, y_k)\}$$
$$= \operatorname*{argmin}_{(x_d, y_d) \in \mathcal{D}} \text{top-k} \, \|x_t - x_d\|_2 \quad (1)$$

where $x_t$ is a neural encoding of a test/query tweet and $x_d$ is the neural encoding of a train tweet from train datastore $\mathcal{D}$. Both $x_d$ and $y_d$, which represents one (of the potentially many) associated tweet's hashtag labels, are stored "unrolled" in $\mathcal{D}$, such that tweets with $h$ hashtags occur $h$ times with only one hashtag per entry. After obtaining the top-$K$ tweet-hashtag pairs $\mathcal{S}_k(x_t)$, we add a final re-ranking step to return final hashtag predictions $\hat{\mathcal{Y}}_r$, where $r << k$:

$$\hat{\mathcal{Y}}_r = \{\hat{y}_1, \dots, \hat{y}_r\} = \text{re-rank}(\mathcal{S}_k(x_t)) \quad (2)$$

Based on this, our dense KNN model has three main components: (1) A *static* neural tweet encoder, such as the fine-tuned BART model (Lewis et al., 2020) that we use, which is only trained once and does not require updates, (2) a datastore that enables fast nearest neighbor search, and (3) a re-ranker to boost recall performance. We explain components (2) and (3) in more detail next.

**Datastore and Search.** We follow prior work (Khandelwal et al., 2020, 2021) and use FAISS (Johnson et al., 2019), which enables efficient indexing, clustering, and approximate nearest neighbor similarity search for dense vectors using quantization, which is faster than exact search, but does not precisely correspond to exact L2 distance. See § 4.1 for implementation details.

**Re-ranking Retrieved Hashtags.** For optimal performance, we explore several methods for re-ranking the top-$K$ retrieved tweets to return the "unique" top-$R$ ($R << K$) hashtags, after removing repetitions. First, **default distance ranking** involves re-ranking based on tweet embedding L2-distance returned by FAISS, which are approximate as FAISS quantizes vectors for search efficiency. Second, our **actual distance ranking** method uses the actual distance between the encoded query tweet and its retrieved neighbors. Third, **frequency-based ranking** prioritizes the

$R$ hashtags with the greatest support in the initial retrieved $K$. We count the number of occurrences of each hashtag in the top-$K$ retrieved tags, and then rank them from most repeated to least, and return the top-$R$ most common ones.

## 3 Longitudinal Hashtag Dataset

Since we require a dataset with fine-grained temporal annotations and aim to study longitudinal effects of temporal distribution shift, we collect a new large-scale benchmark dataset for temporal hashtag prediction on Twitter. We scrape tweets from the entire 2021 calendar year organized by week published, keeping only tweet-hashtags in the top-$10K$ most frequent hashtags that week. We observe large label distribution shift over time, as roughly $50\%$ of labels are new after 4–5 months (for detail, see Fig. 3).

For temporal evaluation, we divide our dataset into alternating 3-week train/1-week test splits (nearly, but not exactly aligned with month boundaries), such that the test data is always from a future date. We chose to divide our dataset into weeks because we (a) needed a fine-grained enough division to investigate how model performance decays over time, yet (b) enough data points in each division for supervised training/fine-tuning of high-capacity neural baselines for comparison. Of course, these two decisions are at odds with each other since very fine-grained divisions will reduce the amount of training data in each division. Finally, (c) we desired a temporal setup that would be practical for real-world use (i.e. avoid excessively frequent updates). For comparison, we also consider the setting with no temporal shift, where each week contains its own train/test splits and all of the 3-week test splits are aggregated for evaluation. To facilitate models like ours that require a frozen historical training corpus (e.g. to train our static tweet encoder), we reserve the first 3-week time split.

## 4 Experimental Setup

For the parametric **neural classifier**, we use BART-large (Lewis et al., 2020) with a multi-label classification head. To explore if performance degradation over time is simply due to a fixed label set, we provide another parametric model, a **neural seq2seq** baseline based on BART with conditional generation head, which can generate hashtags unseen during training. The model is trained to maximize the

---

Code/data (tweet IDs) available here: https://github.com/mireshghallah/temporal-knn

conditional probability of the tokenized sequence of concatenated hashtags, given the tweet text as input. During test time, we force the model to generate 120 tokens, and then select the first 5 hashtags for calculating recall independent of their order.

For our model's **neural encoder for KNN**, we use the same BART architecture as a tweet text encoder. In this setup, we use the neural classifier fine-tuned *once* on the initial 3-week training set for hashtag classification, and then *freeze and re-use* it for encoding every subsequent train/test week in the year. That is, instead of continuously re-training our neural encoder for KNN, we simply re-use it and swap out the encoded datastores during updating. We describe other encoder variants and their results in §5.

### 4.1 Training Details

For the neural classifier experiments, we train each classifier for 30 epochs, and choose the best checkpoint based on validation recall. We use a learning rate of $3e$-5 with a polynomial scheduler and training batch size of 36 for both the neural classifier and the seq2seq models. For the Tw/oA setting introduced in Section 5, we train one model each for classifier/seq2seq. For the classifier, this results in a fixed label set size of 16,886 for all test sets. For the Tw/A setting, we train 12 different models on each of the 3-week alternating train splits, testing on each of the 1-week future test sets with the most recent models.

For the datastores, we encode all tweets into an $N \times E$ memory maps, where $N$ is the number of unrolled tweets (§2) in the training set and $E$ is the embedding size ($E = 1024$). Separately, we construct an $N \times V$ memory map for hashtags, where $V = 280$ is the max allowable length of tokens. We use the FAISS `IndexFlatIP` quantizer with $L2$ as the distance metric. For retrieval, we use their `search` function as well, which retrieves the nearest $K$ neighbors but using the approximate (quantized) distances which means the retrieval and order is not exact. We also experimented with Cosine distance and found it to under-perform compared to $L2$ in our case. We use $K = 1204$ for our experiments (see App. Table 3).

## 5 Results

Our experimental results consist of performance comparisons between (1) different models and temporal settings and (2) user-deletion. We present model ablations in Appendix A.

**Temporal Adaptation.** We perform experiments with three different temporal dataset (§3) settings: (1) **Non-temporal (NT)**, where there is *no* temporal distribution shift between train and test; each week contains its own train/test splits and the 3-week test splits that correspond with each 3-week train split are aggregated for evaluation. (2) **Temporal w/o Adaptation (Tw/oA)**, where a model's training data consists of only the *first* 3-weeks of the year and its test data is from every later test week (weeks 4, 8, ..., 48). (3) **Temporal w/ Adaptation (Tw/A)**, where the model is re-trained throughout the year on the 3-week split immediately before the given test week (e.g. for test week 8, the model is trained on weeks 5–7).

In Table 2 (a), we summarize results across these 3 settings. In terms of comparing the two parametric baselines, we can see that in the temporal setups (Tw/oA, Tw/A) the Seq2Seq model outperforms the Classifier, potentially due to its capability of generating unseen hashtags, unlike the Classifier's fixed training label set. The Classifier is able to perform well exclusively for the NT evaluation, which does not exhibit any distribution shift, and therefore is not a representative setting for production. Dense KNN outperforms Seq2Seq and Classifier for both temporal setups, which shows how simple, yet robust non-parametric models are under distribution shift.

In Fig 2 (b), we compare year-long performance degradation of Tw/oA & Tw/A models for a better understanding of the effects of temporal shift. As the test week progresses further along in the year, we see a marked decrease in performance (as much as a 34% relative decrease) for Tw/oA models (dashed line), which is indicative of major distribution shift between the old data the model was trained on vs. the new test data. Alternatively, the solid lines, representing the continuously re-trained models with the Tw/A setting, tend to maintain their performance much better. Our dense KNN model performs the best across time, despite being the simplest and most efficient model to update since it requires no gradient-based training.

**Does distribution shift affect the tweet encoder?** While our neural tweet encoder—the component of our dense KNN model that encodes tweets as keys for our datastores—requires historical data to learn how to embed tweets for nearest hashtag retrieval, results in Fig 2 (b) show that this older data has no impact on the performance of our full "Dense KNN" model over time. To understand why our approach does not suffer performance degradation despite using a tweet encoder trained on stale data, we can compare our "Dense KNN Tw/A" results in Fig 2 (b) with the "Classifier Tw/oA" setting. As a reminder, these two methods share the same underlying BART encoder model (except for the classification head), which is trained on the first three-week training dataset. While the neural encoder stays the same for "Dense KNN Tw/A" and "Classifier Tw/oA" settings, the "Dense KNN Tw/A" model gets updated with the temporally closest datastores over time, while the "Classifier Tw/oA" must only rely on the stale neural encoder/classifier head. If we contrast "Dense KNN Tw/A"'s purple solid line in Fig 2 (b) with the "Classifier Tw/oA"'s orange dashed line, we observe a sharp performance drop for the stale classifier. However, performance stays relatively constant over time for the "Dense KNN Tw/A" model despite having the same encoder as the stale classifier. We understand this to show that the learned representations for comparing tweets, the component shared by both methods, are not negatively impacted by the passage of time, but it is instead the classification head that becomes outdated as a result of changing label set distributions. Since "Dense KNN Tw/A" is able to see the most recent, empirical label set distribution when we update the datastore over time, this would explain its high performance. These findings are further supported by the increase in performance from the "Classifier Tw/oA" (dashed orange line) to "Classifier Tw/A" (solid orange line) settings, the "Lifetime of a Hashtag" label set overlap over time (App. Fig 3), and App. Fig 4, in which we show the yearlong hashtag prediction performance breakdown for the best temporally adapted settings.

**Deletion Adaptation.** As explained in §1, efficient model adaptation facilitates selective deletion of training data, which can be an important feature for removing harmful training examples or obeying user deletion requests. For non-parametric methods like dense KNN, deletion involves straightforward removal of the given tweet from the datastore, followed by a re-indexing operation, which takes minutes on CPU. The parametric classifier, however, needs to be fully re-trained without the deleted tweets, which takes about a week on our dataset with a single A100 GPU. In Fig 2 (c), we compare our dense KNN with training data from weeks 9–11 and test data from future weeks 12,

|  | Non-Temporal | | Temporal w/o Adapt | | Temporal w/ Adapt | |
|---|---|---|---|---|---|---|
| Method | @5 | @1 | @5 | @1 | @5 | @1 |
| MC Train | 1.7 | 0.5 | 1.1 | 0.3 | 1.7 | 0.5 |
| Classifier | **39.9** | **14.8** | 14.8 | 5.4 | 23.3 | 8.3 |
| Seq2Seq | 34.4 | 13.0 | 16.0 | 6.1 | 23.5 | 8.3 |
| **Dense KNN** | 39.5 | 13.2 | 18.5$^{\dagger}$ | 6.7$^{\dagger}$ | 26.2$^{\dagger}$ | 9.6$^{\dagger}$ |

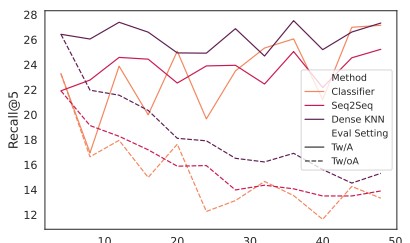 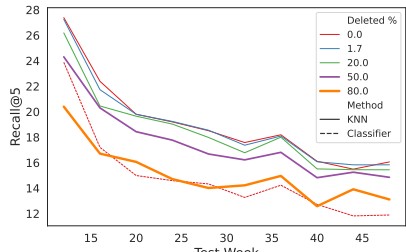

Figure 2: **(a, left)** Comparing average test recall@$\{1, 5\}$ performance over the year. "Most Common Train" (MC Train) uses the most frequent hashtag label from the corresponding training set for each setting. For instance, "Non-temporal" and "Temporal w/ Adaptation" settings use the most common hashtag label from the three weeks right before the test week, unlike "Temporal w/o Adaptation", which uses the most common hashtag label from the first three weeks of the year. $^{\dagger}$ denotes statistical significance via paired bootstrap test (Koehn, 2004). **(b, center)** Comparison of our KNN approach with the neural seq2seq and classifier models. Dashed lines show evaluation results of a model trained/datastore created on only time bucket 1 (first 3 weeks). Solid lines show the performance of adapted/re-trained models on the most recent data. **(c, right)** Performance effects of training data deletion (thicker line means higher deletion percentage) under Temporal w/o Adaptation (Tw/oA) setting. We have plotted the optimal classifier, without any deletion, as its the upperbound of classifier performance.

16, 20, ..., 48 under 4 different training dataset deletion percentages: 1.7%, 20%, 50% and 80%. The 1.7% scenario represents the real-world user-deleted data percentage (due to account suspension, user deletion, or content violations) that is obtained after re-scraping the tweets 5 months after initial collection, while other percentages are chosen for interest and deleted randomly. We compare the behavior of dense KNN with updated datastores with that of the parametric classifier without any deletion in Fig 2 (c). We observe that the KNN model decays gracefully as more data is deleted, and even with the deletion, it still outperforms the upper bound classifier model.

## 6 Related Work

**Hashtag Recommendation.** Earlier works on hashtag recommendation compute tf-idf vectors from extracted keywords and apply classifiers on these vectors (Jeon et al., 2014; Sedhai and Sun, 2014), while late works incorporate neural nets (Li et al., 2016; Shen et al., 2019; Gong and Zhang, 2016; Ma et al., 2019). Feng and Wang (2014) collected a dataset of millions of tweets/hashtags across two months of 2012 and designed a personal hashtag recommendation model using user meta-data. Instead, we release a public dataset of tweets over an *entire year*, show how to continually update a non-parametric datastore to *handle temporal shift*. Another line of research models semantic word shift as one form of temporal change (Wijaya and Yeniterzi, 2011; Kulkarni et al., 2015; Hamilton et al., 2016; Kutuzov et al., 2018).

**Temporal Adaptation.** More recently, the temporal generalization problem has been re-emphasized in pretrained language models (Lazaridou et al., 2021; Dhingra et al., 2022; Luu et al., 2022; Jin et al., 2022; Loureiro et al., 2022) as their re-training cost continues to grow. While in-context learning enables updating model posteriors without gradient updates, static models are still unable to update their temporal world knowledge (Akyürek et al., 2022; Xie et al., 2021). The temporal distribution shift problem on Twitter has been studied in Preoţiuc-Pietro and Cohn (2013); Rijhwani and Preotiuc-Pietro (2020); Luu et al. (2022); Kowald et al. (2017); Kamath and Caverlee (2013).

**User Data Deletion.** While several methods have been proposed to partially mitigate computational costs of unlearning through techniques to speed up re-training, substantial compute is still required (Bourtoule et al., 2021; Wu et al., 2020). Decremental learning (Cauwenberghs and Poggio, 2000; Karasuyama and Takeuchi, 2009) has been proposed for removing specific training samples. Different from previous methods, we explore a simple, theoretically guaranteed, and effective way to delete user data by modifying the datastore.

## 7 Conclusion

In this paper, we show that a non-parametric dense KNN retriever model is particularly well-suited for the task of temporal adaptation as it can efficiently adapt to changing test distributions over time by easily updating and swapping out datastores. We find it improves by 64% over the best static parametric baselines while avoiding their costly gradient-based re-training. Our model also performs well under selective deletion of training data, which is an important feature for removing harmful training examples, or obeying user deletion requests in line with data privacy laws, such as GDPR and CCPA. We hope the release of our dataset will encourage more work on temporal model adaptation in the future.

## Limitations

Our proposed dense retrieval method relies on a collection of "historic data" which is used to train an encoding/embedding model, to produce the representations that would be used as keys in the datastore. As the KNN method relies on these representations for finding neighbors, it is crucial to train the embeddings on in-domain historic data, that would also be unlikely to receive deletion requests. However, as we show in our experiments, this data does not go stale for even a year out, and still performs well and provided appropriate embeddings, as the performance of the KNN method does not degrade with the passage of time (as seen in Fig 2 (b) and discussed in Section 5).

## Ethics Statement

We abide by the Twitter development and data usage agreement (https://developer.twitter.com/en/developer-terms/agreement-and-policy) for the collection and usage of the Twitter data. Also, upon release of the dataset, we will only make the tweet IDs available for data re-hydration (and not the user ID or the tweet body), to protect the privacy of the users, which is inline with the overall goal of this paper as well.

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

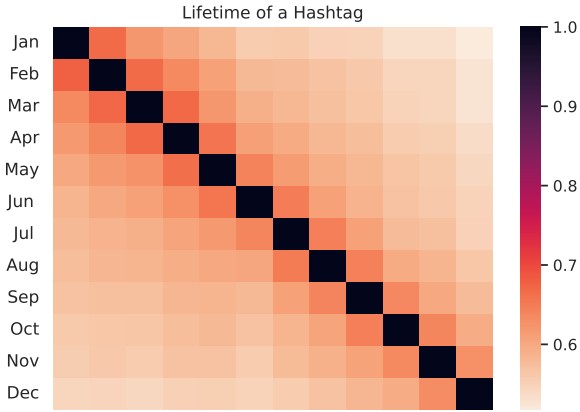

Figure 3: On Twitter, hashtag usage exhibits significant temporal distribution shift, which is challenging for current NLP models. We show the hashtag label set overlap, computed as recall between each month's hashtags, for our longitudinal dataset of 7.13M tweets over the 2021 calendar year. We note that the recall is not symmetric because monthly hashtag sets have different sizes.

| Longitudinal Hashtag Dataset Statistics | |
|---|---|
| Number of Train Tweets Per Week | 475,437 |
| Number of Val-Test Tweets per Week | 59,430 |
| Number of Avg Tags per Tweet | 2.9 |
| Number of Unique Tags per Week | 10,000 |
| Avg Hashtag Length (tokens) | 3.2 |
| Avg Tweet Length (tokens) | 30.3 |

Table 1: Summary statistics of new longitudinal dataset.

## A  Ablation Studies

In this section, we perform an ablation of several model components, including the encoding of the tweets for building the datastore, the $K$ in KNN, and the re-ranking of the retrieved $K$ nearest neighbors. Finally, we break-down the recall @5 results of the KNN over different time buckets, to see how well the updated datastore helps capture out-of-vocabulary tags that a datastore/model from the first time bucket would not have captured.

**Ablating encoder for datastore.**  Apart from the classifier encoder used in all previous experiments, we also tried using both the seq2seq model as the encoder, and a generic encoder trained on tweets, named Bertweet (Nguyen et al., 2020), compared in Table 2. The classifier encoder performs the best, followed by the seq2seq, and Bertweet. One potential explanation for Bertweet's poor performance is that its training set, which consists of tweets from 2012 to 2019, is outdated, creating a significant distribution mismatch. This would hint that the KNN

| | Non-temporal | | Temporal | | | |
|---|---|---|---|---|---|---|
| | | | W/o Adaptation | | W/ Adaptation | |
| | @5 | @1 | @5 | @1 | @5 | @1 |
| MC Train | 1.69 | 0.46 | 1.11 | 0.26 | 1.68 | 0.47 |
| KNN-Seq | 35.21 | 11.83 | 10.35 | 3.32 | 19.64 | 5.88 |
| KNN-Bertweet | 33.07 | 11.00 | 7.55 | 2.30 | 15.46 | 4.59 |
| **KNN-Clsf** | **39.54** | **13.15** | **18.45** | **6.65** | **26.21** | **9.60** |

Table 2: Ablation study of the effect of different encoders for encoding the tweets in the datastore and retrieving the nearest neighbors.

encoder could also require updates, although only across years as previously described.

**Ablating $K$ and re-ranking methods.**  In Table 3, we show ablation results for top-$k$ and re-ranking averaged over the 12 test weeks. Overall, frequency-based re-ranking outperforms the distance-based methods by a large margin, which we hypothesize is due to the robustness added by the repetitions in hashtags. We can see that for the distance-based methods, the overall trend is that higher $K$ is better, due to fewer cascading errors (1024 is optimal on average). The frequency-based re-ranking, however, degrades significantly if the number of retrieved neighbors is large (1024 and 2048), which is expected, as more irrelevant but common hashtags are suggested. When $K$ approaches the datastore size, we approach a random frequency-based classifier.

**OOV tag prediction performance.**  Finally, we want to investigate how updating the datastore helps us capture new, out-of-vocabulary (OOV) hashtags, that would not be predicted if we kept using the datastore from time bucket 1. Fig 5 shows the results for this experiment, where the OOV refers to out-of-vocabulary with respect to bucket 1's hashtag vocabulary. IV refers to in-vocabulary hashtags, which means the tags that appear both in the given test week, and the train data of time bucket 1. We report the recall over the IV and OOV tags separately. We can see that the updates in the datastores help predict $19\%$ of the hashtags that would otherwise not be predicted, on average across the test weeks. We see that as we proceed with time buckets, the OOV recall grows, eventually overtaking the IV recall. It is worth noting that the number of IV tags is substantially smaller in later time buckets. We suspect the superior OOV performance on the later time buckets is related to content shift, i.e., the meanings of IV tags may have shifted by the later time buckets.

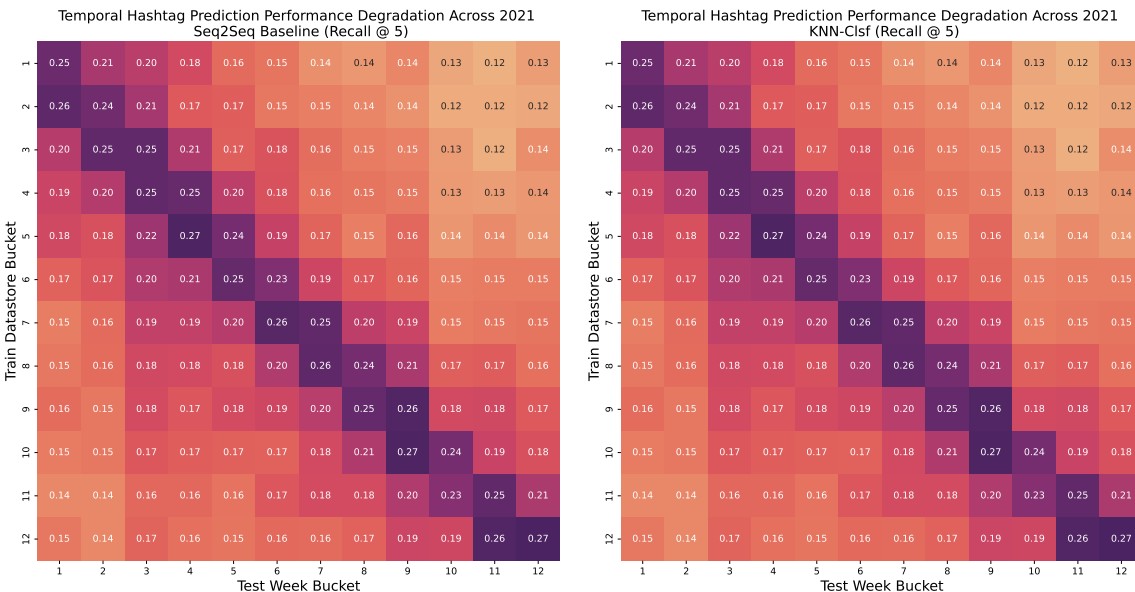

Figure 4: Evaluation matrix using all possible combinations of previous three-week-span train buckets (roughly corresponding to months) to predict hashtags for all possible combinations of test week buckets (occurring directly after each three-week-span train bucket throughout the year). We compare recall@5 results for the best performing seq2seq temporal baseline from Table 2 (a), and our Dense KNN model. We include using future months to predict past test weeks, although unrealistic in practice, for completeness.

| Method / K | 20 | 50 | 100 | 1024 | 2048 |
|---|---|---|---|---|---|
| **Frequency-based** | **25.23** | **26.26** | **26.21** | 23.12 | 21.44 |
| Default Dist. | 24.00 | 24.09 | 24.32 | 24.27 | 24.21 |
| Actual Dist. | 24.22 | 24.50 | 24.77 | **24.97** | **24.92** |

Table 3: Effect of $K$ on KNN retrieval and comparing different re-ranking methods from §2. Recall @ 5 is reported, averaged over the 12 test weeks of the year under the Tw/A evaluation scheme.

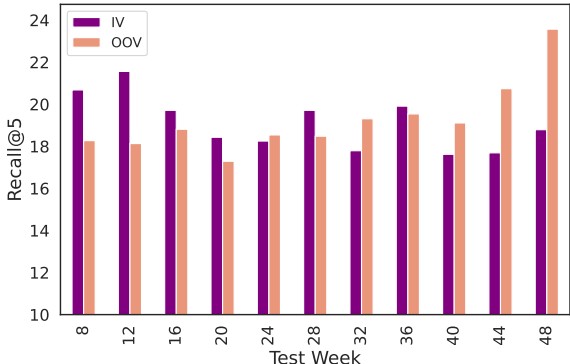

Figure 5: Studying the effect of updating the datastore on capturing out-of-vocabulary (OOV) vs. in-vocabulary (IV) hashtags. OOV recall is reported with respect to the vocabulary from the initial three-week training set.