# OpenReview forum: "Simple Temporal Adaptation to Changing Label Sets: Hashtag Prediction via Dense KNN"
_EMNLP/2023/Conference — EMNLP 2023 Main_

### Official Review · Reviewer_9dfY · 2023-08-03

**Soundness:** 3

**Excitement:**

3: Ambivalent: It has merits (e.g., it reports state-of-the-art results, the idea is nice), but there are key weaknesses (e.g., it describes incremental work), and it can significantly benefit from another round of revision. However, I won't object to accepting it if my co-reviewers champion it.

**Paper Topic And Main Contributions:**

This paper addresses a challenge in NLP related to the adaptability of traditional models to the constantly changing nature of user-generated social media data. By focusing on temporal adaptation through longitudinal hashtag prediction, the authors propose a non-parametric dense retrieval technique to handle temporal distribution shifts and deleted training data without the need for costly re-training. The paper presents an approach with practical implications. However, some aspects deserve further attention and refinement.

**Reasons To Accept:**

1. The paper introduces a non-parametric dense retrieval technique that offers a simple and effective solution to adapt NLP models to temporal changes.
2. The experiments conducted on a year-long Twitter dataset demonstrate an improvement over the baselines.
3. The paper's focus on user-generated social media data and its adaptability challenges is relevant in the current digital landscape.


**Reasons To Reject:**

1.	Lack of detailed description of the dataset in the main sections, such as data volume.
2.	What is the basis for setting time division?
3.	My understanding is that the author classifies new tweets into the static hashtag of the training set, but the hashtag itself changes with time. How does the author deal with this problem?

**Reproducibility:**

3: Could reproduce the results with some difficulty. The settings of parameters are underspecified or subjectively determined; the training/evaluation data are not widely available.

**Reviewer Confidence:**

3: Pretty sure, but there's a chance I missed something. Although I have a good feel for this area in general, I did not carefully check the paper's details, e.g., the math, experimental design, or novelty.

---

> ### Author Rebuttal · Authors · 2023-08-29
>
> Thank you for your time and helpful feedback on our submission. We hope to address your questions and concerns below:
>
>
> **1.** We appreciate the suggestion to include a more detailed dataset description in the paper. To address this, we will (a) move our dataset statistics table that is currently in the Appendix into the main body of the paper in all future revisions, and (b) elaborate on our dataset construction and provide examples of its contents in the main paper, which we were unable to include due to the short paper format’s space constraints. We hope that this will encourage others to use the dataset for future research on temporal model adaptation.
>
>
> **2.** In addition to providing more details on the dataset in future revisions, we will also elaborate on how we decided the temporal division. In order to test temporal effects at-scale, we chose to divide our dataset into weeks because we (a) needed a fine-grained enough division to investigate how model performance decays over time, yet (b) enough data points in each division for supervised training/fine-tuning of high-capacity neural baselines for comparison. Of course, these two decisions are at odds with each other since very fine-grained divisions will reduce the amount of training data in each division. Finally, (c) we desired a temporal setup that would be practical for real-world use (i.e. avoid excessively frequent updates). While, in our paper, we showed that updating a model every three weeks provides noticeable gains in performance, we note that due to the hashtag prediction task’s inherent user-generated labels, more fine-grained (i.e., day-to-day) splits could just as easily be gathered in an industrial setting for a production system to further reduce temporal label shift.
>
>
> **3.** Good question – this is a key feature of our approach. Because our proposed model (Dense KNN w/ temporal adaptation) is non-parametric, temporal updates can be performed simply by replacing the datastore with the most recent batch of tweets and hashtags. As a side effect, replacing the datastore updates our hashtag label inventory: a dense KNN retriever’s label set is fully determined by its datastore. The BART Classifier baseline can only predict the tags seen in the training data as well due to the fixed output layer size, but requires costly gradient updates to do so. This is because the BART Classifier is parametric: the number of parameters is fixed when it is trained, so new hashtag labels require new parameters.  Finally, in order to have a complete experimental comparison, we wanted to also include a parametric baseline that had at least the possibility of predicting new hashtags. For this purpose, we included the BART seq2seq baseline. Because it generates hashtags token by token, this baseline has an open-vocabulary: it has the ability to place probability mass on hashtags it never saw during initial training. In our experimental results, we show that neither the Classifier nor the Seq2seq model perform as well as our temporally updated Dense KNN retrieval model, *even when they are temporally updated with new supervised training data.* This demonstrates that our Dense KNN model is an effective approach to label shift in temporal classification despite requiring no costly gradient-based re-training like the Classifier/Seq2seq model. Its efficiency also suggests that it could be automatically updated much more often than other methods (i.e., at daily intervals) to reduce the occurrence of unseen hashtags.

---

### Official Review · Reviewer_aHrb · 2023-08-03

**Soundness:** 3

**Excitement:**

3: Ambivalent: It has merits (e.g., it reports state-of-the-art results, the idea is nice), but there are key weaknesses (e.g., it describes incremental work), and it can significantly benefit from another round of revision. However, I won't object to accepting it if my co-reviewers champion it.

**Paper Topic And Main Contributions:**

The authors' contribution lies in introducing a novel task that focuses on predicting hashtags for tweets using temporal adaptation, taking into consideration the real-time nature of social media. Traditional models trained solely on historical data might not be suitable for handling new data in this context. To address this challenge, the authors propose a method that leverages a constructed datastore to retrieve similar tweets, thus providing valuable contextual information to the model. Consequently, the proposed model demonstrates the ability to predict hashtags for new tweets without the need for re-training. The evaluation of their approach is conducted using tweet data gathered from Twitter, and the dataset is thoughtfully partitioned into weekly segments to perform comprehensive experiments.

**Questions For The Authors:**

A. What does the “MC Train” in Figure 2 mean?

**Reasons To Accept:**

1. The paper introduces a new task focusing that revolves around hashtag prediction, considering the impact of temporal shifts.
2. The authors propose a retrieval-based method that enables the model to predict hashtags without the burdensome requirement of re-training.

**Reasons To Reject:**

1. The experimental results derived from the collected dataset fail to convincingly demonstrate the universal effectiveness of the proposed method. Specifically, as indicated in Table 2, the performance of KNN-Seq and KNN-Bertweet with adaptation exhibits a lower recall @1 compared to their counterparts without adaptation, raising concerns about the method's reliability and applicability.

2. The decision to split the data based on a weekly interval for real-world scenario evaluation appears somewhat impractical, as it results in data sets with closely aligned training and testing dates. To enhance the validity of the experiments, it is suggested to adopt a more reasonable time interval, such as training on data from January, February, and March, while testing on data from July, to simulate a more realistic temporal separation.

3. The paper lacks a significant test to ascertain the statistical significance of the reported results.

**Reproducibility:**

3: Could reproduce the results with some difficulty. The settings of parameters are underspecified or subjectively determined; the training/evaluation data are not widely available.

**Reviewer Confidence:**

3: Pretty sure, but there's a chance I missed something. Although I have a good feel for this area in general, I did not carefully check the paper's details, e.g., the math, experimental design, or novelty.

---

> ### Author Rebuttal · Authors · 2023-08-29
>
> Thank you for your time and helpful feedback on our submission. We hope to address your questions and concerns below. In summary, we’ve fixed the error you pointed out in our ablation table (we had accidentally swapped two columns, which led to the strange results you pointed out), added a full temporal matrix of train/test results for our best method vs. the best baseline method, and performed significance testing to measure the statistical reliability of our results.
>
>
> **Reasons to Reject:**
>
>
> **1.** Thanks so much for noticing this discrepancy in the encoder ablation results in Table 2. We checked our results and found that this was actually a data entry typo due to transferring our ablation results from the model output into the table. Specifically, the recall@1 columns for “W/o Adaptation” and “W/ Adaptation” are actually swapped for the KNN-Seq row and the KNN-Bertweet row. While double-checking this ablation table against the main results table in Figure 2 (left), we also found that the “Frequency baseline” result in Table 2 had its numbers swapped as well. In particular, both the recall@5 and recall@1 columns were swapped for the “W/o Adaptation” and “W/ Adaptation” settings. Here’s the updated table formatted in Markdown with these **fixes marked in bold**:
>
> | Method | NT R@5 | NT R@1 | Tw/oA R@5 | Two/A R@1 | Tw/A R@5 | Tw/A R@1 |
> | :--- | :---: | :---: |  :---: | :---: | :---: | :---: |
> | Frequency baseline | 1.69 | 0.46 | **1.11** | **0.26** | **1.68** | **0.47** |
> | KNN-Seq | 35.21 | 11.83 | 10.35 | **3.32** |19.64 | **5.88** |
> | KNN-Bertweet | 33.07 | 11.00 | 7.55 | **2.30** | 15.46 | **4.59** |
> | KNN-Clsf | 39.54 | 13.15 | 18.45 | 6.65 | 26.21 | 9.60 |
>
>
> After fixing this data entry error, the recall@1 numbers now have better agreement with the recall@5 settings across each row. Also, the “Frequency baseline” is correct now, since it’s actually the same baseline as the “MC Train” frequency baseline from Figure 2 (left) but with an extra digit of precision. We would like to emphasize that these numbers are from our additional experimental ablation table added for completeness in the Appendix Table 2 and not from our main results in the main paper (located in Figure 2 (left)). We apologize for the confusion, and will release all scripts to verify these results along with the other code/data.
>
>
> **2.** Thanks for your suggestion, we think a full evaluation result matrix would be a great addition to the paper. For this rebuttal, we performed an exhaustive evaluation setup using all possible combinations of previous three-week-span train buckets (roughly corresponding to months) to predict hashtags for all possible combinations of future test week buckets (occurring directly after each three-week-span train bucket throughout the year). We compare recall@5 results for the best performing temporal baseline from Figure 2 (Seq2seq), and our Dense KNN model in separate 12x12 Markdown tables below:
>
> **Seq2seq w/ Adaptation Baseline Results**
>
>
> |Train Datastore Bucket/Test Week Bucket|  4  |  8  |  12  |  16  |  20  |  24  |  28  |  32  |  36  | 40  | 44  | 48  |
> |------------------------------:|----:|----:|----:|----:|----:|----:|----:|----:|----:|----:|----:|----:|
> |                              **1**|18.7|16.9|16.7|14.4|17.1|16.8|14.4|15.3|13.9|14.8|15.4|14.3|
> |                              **2**||21.0|20.3|15.6|17.1|16.8|13.2|17.6|14.6|13.0|15.4|12.8|
> |                              **3**|||25.3|16.8|19.1|18.4|14.0|18.6|16.6|15.0|16.7|13.8|
> |                              **4**||||21.7|20.9|19.8|18.2|15.8|17.0|14.1|14.9|13.5|
> |                              **5**|||||23.9|20.1|16.3|17.8|17.4|17.9|17.8|17.4|
> |                              **6**||||||22.7|17.8|17.1|17.9|15.9|17.6|16.0|
> |                              **7**|||||||24.1|19.1|19.7|15.3|18.1|16.4|
> |                              **8**||||||||22.4|22.9|17.5|19.3|17.0|
> |                              **9**|||||||||26.9|19.0|21.0|17.9|
> |                             **10**||||||||||21.2|23.2|18.6|
> |                             **11**|||||||||||27.8|21.1|
> |                             **12**||||||||||||26.4|
>
>
>
> **Dense KNN w/ Adaptation Results**
>
>
> |Train Datastore Bucket/Test Week Bucket|  4  |  8  |  12  |  16  |  20  |  24  |  28  |  32  |  36  | 40  | 44  | 48  |
> |------------------------------:|----:|----:|----:|----:|----:|----:|----:|----:|----:|----:|----:|----:|
> |                              **1**|24.7|21.0|19.9|18.3|16.1|15.5|14.0|13.5|14.5|12.6|12.0|13.1|
> |                              **2**||24.4|21.2|17.2|16.9|15.3|14.9|14.5|13.9|12.3|12.3|12.5|
> |                              **3**|||25.4|20.6|16.8|17.6|15.9|15.1|14.8|13.2|12.1|14.1|
> |                              **4**||||25.0|19.8|17.7|15.8|15.5|15.2|13.3|13.1|13.5|
> |                              **5**|||||24.1|19.2|16.5|14.9|16.3|13.6|14.2|13.6|
> |                              **6**||||||23.5|18.7|17.2|16.4|14.9|15.1|15.1|
> |                              **7**|||||||25.4|19.7|18.9|15.1|15.3|15.5|
> |                              **8**||||||||24.1|20.9|16.8|17.0|15.8|
> |                              **9**|||||||||26.3|18.3|18.4|17.2|
> |                             **10**||||||||||24.1|19.0|17.8|
> |                             **11**|||||||||||25.3|20.8|
> |                             **12**||||||||||||26.8|
>
>
> In these results, we can get a better understanding of how both methods generalize to longer time horizons. We plan to include all results in this format in future revisions.
>
> **3.** Thank you for pointing this out – we agree that significance testing is needed in order to establish that these results are generalizable. In future revisions, we will add confidence intervals for systems, and paired tests for the most important comparisons. For time and space during rebuttal, we report the results of a paired significance test for the most critical comparison: the comparison of the average performance of our best model (Dense KNN w/ Adaptation) with that of the next best model (Seq2seq w/ Adaptation). We performed a paired bootstrap that resampled the test set 50,000 times with replacement ([Statistical Significance Tests for Machine Translation Evaluation](https://aclanthology.org/W04-3250); Koehn, EMNLP 2004). We find that the gain of Dense KNN over Seq2seq is statistically significant at the p<0.001 level – indicating that the superior performance of our proposed model is likely statistically reliable. Further, since all other baselines perform worse than Seq2seq, we have every expectation that our approach has statistically significant gains over the other baselines as well.
>
> **Questions:**
>
>
> **A.** “MC Train” is an abbreviation for “Most Common Train”, which refers to a baseline where we return the most frequent hashtag label from the corresponding training set for each setting. For instance, in the “Non-temporal” and “Temporal w/ Adaptation” settings, we always return the most common hashtag label from the three weeks right before the test week. For the “Temporal w/o Adaptation” setting, we always return the most common hashtag label from the first three weeks of the year. We included this baseline to show what performance looks like for a method that relies simply on common hashtag usage statistics to recommend hashtags to users. We chose this “most common” approach because it performed best for our Dense KNN re-ranker. However, we realize we did not include an in-depth explanation of this and will do so in future versions of the paper. Thanks for pointing this out.

---

### Official Review · Reviewer_sW7e · 2023-08-04

**Soundness:** 4

**Excitement:**

4: Strong: This paper deepens the understanding of some phenomenon or lowers the barriers to an existing research direction.

**Paper Topic And Main Contributions:**

Challenges: the high cost of maintaining training data for model retraining over time. User-generated social-media data is influenced by new trends that shape online discussions and altered by deleting personal information for privacy considerations. Traditional NLP models rely on fixed training datasets and cannot adapt to time variations. Most of the current research explores the adaptation of traditional fields and requires retraining models.

Contributions: The authors investigate temporal adaptation through a task of longitudinal hashtag prediction, introducing a dense non-parametric retrieval technique that eliminates the need for retraining. This method improves by 64% recall compared to previous methods and avoids the cost of retraining models.

Contribution types: approaches for data- and compute efficiency.

**Questions For The Authors:**

A. If a new Tweet significantly differs from historical Tweet data, or if the content of the historical data is unrelated to the retrieval content, how will the model rerank them and infer this situation?

B. We suggest the authors provide reasons or explanations for the proposed approach, such as why they chose the KNN model over other models, dataset preparation, and illustrations for parameter settings, like top-k and top-R.

**Reasons To Accept:**

Critical issue: social-media data is continually increasing, changing, and shifting, making the cost of annotating or maintaining such data over time relatively expensive. Most current research explores the technology of supervised learning models, but the cost of labeling datasets and retraining models needs to be considered.

Novel ideas: this paper applies the dense KNN retrieval model to reduce the cost of model retraining and maintain the model's adaptability based on shifts in time distribution.
The proposed method appears to be easy to implement and innovative, showing good performance in the Twitter dataset. We believe that such research, which can reduce annotation costs and reduce the need for retraining models, is worthy of further study for exploration.

**Reasons To Reject:**

Perhaps due to the limitation of a short paper, the authors did not provide sufficient explanations for the proposed method, such as the reason for choosing the KNN model, the parameters setting of top-k and top-R, etc. But we think these are minor weaknesses that can be revised.

**Reproducibility:**

3: Could reproduce the results with some difficulty. The settings of parameters are underspecified or subjectively determined; the training/evaluation data are not widely available.

**Reviewer Confidence:**

3: Pretty sure, but there's a chance I missed something. Although I have a good feel for this area in general, I did not carefully check the paper's details, e.g., the math, experimental design, or novelty.

---

> ### Author Rebuttal · Authors · 2023-08-29
>
> Thank you for your time and helpful feedback on our submission. We hope to answer your questions below.
>
>
> **A.** While our neural tweet encoder---the component of our dense KNN model that encodes tweets as keys for our datastores---requires historical data to learn how to embed tweets for nearest hashtag retrieval, results in Figure 2 show that this older data has no impact on the performance of our full “Dense KNN” model over time. To understand why our approach does not suffer performance degradation despite using a tweet encoder trained on stale data, we can compare our “Dense KNN Tw/A” results in Figure 2 with the “Classifier Tw/oA” setting. As a reminder, these two methods share the same underlying BART encoder model (except for the classification head), which is trained on the first three-week training dataset. While the neural encoder stays the same for “Dense KNN Tw/A” and “Classifier Tw/oA” settings, the “Dense KNN Tw/A” model gets updated with the temporally closest datastores over time, while the “Classifier Tw/oA” must only rely on the stale neural encoder/classifier head. If we contrast “Dense KNN Tw/A”’s purple solid line in Fig 2 (center) with the “Classifier Tw/oA”’s orange dashed line, we observe a sharp performance drop for the stale classifier. However, performance stays relatively constant over time for the “Dense KNN Tw/A” model despite having the same encoder as the stale classifier. We understand this to show that the learned representations for comparing tweets, the component shared by both methods, are not negatively impacted by the passage of time, but it is instead the classification head that becomes outdated as a result of changing label set distributions. Since “Dense KNN Tw/A” is able to see the most recent, empirical label set distribution when we update the datastore over time, this would explain its high performance. These findings are further supported by the increase in performance from the “Classifier Tw/oA” (dashed orange line) to “Classifier Tw/A” (solid orange line) settings, and the “Lifetime of a Hashtag” label set overlap over time presented in Figure 3.
>
>
> **B.** We initially choose the dense KNN model over other model classes for two main reasons: (1) it can efficiently adapt to changing test distributions over time by easily updating and swapping out datastores, and (2) it abides by data privacy laws, such as GDPR, CCPA, etc. by removing individual entries from the training data without costly gradient-based re-training of the model. We find that our results in Figure 2 support these desiderata.
>
>
> Regarding our dataset preparation decisions, we elaborate on our  choices in the rebuttal to Reviewer #3 in response 2. To summarize, for temporal division of the dataset, we decided on 3-week train sets/1-week test sets to balance (a) how fine-grained the divisions could be for detailed temporal analysis with (b) the need for large enough divisions to train supervised neural models. We also required (c) a realistic setup for an industrial hashtag recommendation engine that would not require excessively frequent updates.
> Finally, our parameter settings for top-*K* and top-*R* were determined through grid search on a separate validation set. We show the effect of *K* on different tweet encoders in the Appendix Table 3. We will expand on this and also bring the “Training Details” from the Appendix into the main paper with the extra page afforded to us in the camera ready.

---

### Meta-Review · Area_Chair_hLRM · 2023-09-09

**Recommendation:** 3

**Metareview:**

The paper develops a new approach for tackling the challenge of predicting hashtags for tweets in real-time social media by leveraging temporal adaptation. The method utilizes a constructed datastore to retrieve similar tweets, providing informative context and allowing the model to predict hashtags for new tweets without the need for re-training. The evaluation employs Twitter data partitioned into weekly segments, offering comprehensive insights into their approach's effectiveness. While this paper addresses the crucial issue of adaptability in handling evolving social media data, there are some aspects that could benefit from further attention and refinement, as mentioned in the second review.

The reviewers acknowledge the merits of this work and the authors provided additional experiments to address some of the raised concerns. The reviewer agree that the work is sound and has a decent excitement level.

---

### Decision · Program_Chairs · 2023-10-07

**Decision:**

Accept-Main

**Comment:**

The paper develops a new approach for tackling the challenge of predicting hashtags for tweets in real-time social media by leveraging temporal adaptation. The method utilizes a constructed datastore to retrieve similar tweets, providing informative context and allowing the model to predict hashtags for new tweets without the need for re-training. The evaluation employs Twitter data partitioned into weekly segments, offering comprehensive insights into their approach's effectiveness. While this paper addresses the crucial issue of adaptability in handling evolving social media data, there are some aspects that could benefit from further attention and refinement, as mentioned in the second review.

The reviewers acknowledge the merits of this work and the authors provided additional experiments to address some of the raised concerns. The reviewer agree that the work is sound and has a decent excitement level.